# Novel deep reinforcement learning based collision avoidance approach for path planning of robots in unknown environment

Raed Alharthi[1], Iram Noreen[2]*, Amna Khan[1], Turki Aljrees[1], Zoraiz Riaz[3], Nisreen Innab[4]*

**1** Department of Computer Science and Engineering, University of Hafr Al-Batin, Hafar Al-Batin, Saudi Arabia, **2** Department of Computer Science, Bahria University, Lahore Canoys, Islamabad, Pakistan, **3** Department of Computer Science, University of Sialkot, Sialkot, Pakistan, **4** Department of Computer Science and Information Systems, College of Applied Sciences, AlMaarefa University, Diriyah, Riyadh, Saudi Arabia

* iram.bulc@bahria.edu.pk (IN); Ninnab@um.edu.sa (NI)

**Data Availability Statement:** This research work is based on deep Reinforcement learning in which we have designed Collision Avoidance Approach for

## Abstract

Reinforcement learning is a remarkable aspect of the artificial intelligence field with many applications. Reinforcement learning facilitates learning new tasks based on action and reward principles. Motion planning addresses the navigation problem for robots. Current motion planning approaches lack support for automated, timely responses to the environment. The problem becomes worse in a complex environment cluttered with obstacles. Reinforcement learning can increase the capacity of robotic systems due to the reward system's capability and feedback to the environment. This could help deal with a complex environment. Existing algorithms for path planning are slow, computationally expensive, and less responsive to the environment, which causes late convergence to a solution. Furthermore, they are less efficient for task learning due to post-processing requirements. Reinforcement learning can address these issues using its action feedback and reward policies. This research presents a novel Q-learning-based reinforcement algorithm with deep learning integration. The proposed approach is evaluated in a narrow and cluttered passage environment. Further, improvements in the convergence of reinforcement learning-based motion planning and collision avoidance are addressed. The proposed approach's agent converged in 210th episodes in a cluttered environment and 400th episodes in a narrow passage environment. A state-of-the-art comparison shows that the proposed approach outperformed existing approaches based on the number of turns and convergence of the path by the planner.

## Introduction

Motion planning refers to finding a motion sequence for robots from source to destination [1]. It is a critical and essential feature of the autonomous operation of mobile robots, such as unmanned pursuit, rescue robots, self-driving autonomous vehicles, agriculture robots, extra-terrestrial rovers, and medical assistive robots [1–3]. Motion planning is critical to realize the

Path Planning of Robots in Unknown Environment. No dataset generated or required for this work.

**Funding:** The author(s) received no specific funding for this work.

**Competing interests:** The authors have declared that no competing interests exist.

autonomous operation of mobile robots. Several planner algorithms have been used to solve this problem. Grid-based approaches such as A* [4], D* [5], and MEA* (Memory Efficient A*) [6] have been used for small-scale problems. Whereas, sampling-based approaches like RRT (Rapidly Exploring Random Tree) [6, 7] and RRT*-AB (Rapidly Exploring Random Tree Adjustable Bounds) [8] have shown tremendous performance for high-dimensional complex problems. Furthermore, some heuristic-based evolutionary planners have improved the performance of path planning algorithms in real-world scenarios such as GA (Genetic Algorithm) [9], ACO (Ant Colony Optimization) [10, 11], and PSO (Particle Swarm Optimization) [12].

In the near future, robots are expected to serve in close proximity to humans therefore, maintaining safety during navigation is of critical importance [10, 13, 14]. The rise of the service industry is significantly higher compared to the conventional industrial sector. In this era of Industry 4.0, the major need for a production environment is a flexible and adaptive system while maintaining energy efficiency. Specifically, the motion planning domain requires the ability to deal with different environmental variations [13]. A robot's motion plan needs to refresh its sensors to get new data from the environment, which is computationally expensive as well as energy-consuming for a robot [10, 14]. Extensive work has been performed recently based on environmental feedback and human interaction. However, current motion planning approaches lack support for timely automated responses to the environment in real-time navigation scenarios. Moreover, conventional path planning methods such as grid-based, sampling-based, or heuristic-evolutionary planners are slow, computationally expensive, and less responsive to the environment [15]. Moreover, they often need post-processing due to an inefficient path. Recently, the motion planning domain has focused on using the feedback capability potential of reinforcement learning (RL) to resolve the aforementioned limitations. RL is expedient in addressing these issues due to its action feedback and reward policies.

The main elements in RL normally include an agent, an environment, a reward signal, a policy, and a value function. Its learning structure operates as humans learn and adapt to new circumstances. The agent is not told specifically the behavior for a particular circumstance; rather, it has to find the desired behavior on its claim using trial and error. After each state update, the learning agent is given a scalar reward; the better the transfer, the higher the reward, and it tries to optimize the return over time. RL's significant properties include its trial-and-error nature as well as its long-term return maximization. On the other hand, deep neural networks (DNNs) are at the heart of some of the most important developments in AI and machine learning in recent years, such as self-driving vehicles, image recognition systems [16], speech recognition systems [17], and autonomous robots [18]. Deep learning approaches have gained popularity due to their ability to self-learn and high precision. Recent advances in RL using deep learning have brought new suggestions to address the problem of motion planning. However, these algorithms exhibit the issue of slow convergence. In this regard, this study makes the following contributions

- A deep reinforcement learning technique is presented for motion planning in a complex area cluttered with obstacles. The proposed approach is a less complex deep reinforcement learning approach using Q networks for motion planning.

- The proposed approach is tested in two types of environments: an environment with narrow, cluttered passages and an environment cluttered with structural obstacles.

- Proposed a path with improved convergence which provides an energy-efficient path with fewer turns, consequently generating an efficient path with less energy consumption and path following time.

The rest of the paper is organized as follows: Section describes prominent work in the domain of path planning. Section explains the proposed approach and how it works. Section describes the experimental setup and presents the discussion of the results, followed by the conclusion in Section.

## Related work

Advances in learning-based methods in recent years have facilitated developments in the robotics path planning domain for domestic and commercial robots [19]. Among learning-based techniques, DNN, and deep RL (DRL)-based approaches have gained the attention of the research community due to their applicability in a wide range of applications. DRL does not require enormous design information; rather, it attempts to enhance learning based on rewards by playing out activities. Wu et al. [20] presented the autonomous navigation and obstacle avoidance (ANOA) method for unmanned surface vehicles (USVs) using a deep Q-network. The input of the deep Q-network is pixels positions, i.e., coordinates in the environment input image map and the output of the Q-network is a value function. Guo et al. [21] introduced a long short-term memory (LSTM) based DRL method for path planning of unmanned aerial vehicles (UAVs). The main focus of both approaches was to resolve the fundamental limitation of poor convergence caused by a huge amount of collective data in a high-dimensional dynamic environment.

Semnani et al. [22] presented a hybrid algorithm to solve distributed motion planning in dense and dynamic environments using reinforcement learning and forced-based motion planning (FMP) to address the issues of energy-efficient path generation in feasible time and poor convergence of collision-free path in a cluttered environment. The aforementioned problems were resolved by introducing a goal distance-based proxy award function. Devo et al. [23] proposed a two-step DRL architecture to address the problem of visual path planning in random random-sized three-dimensional (3D) maze environments. The first component translates the commands to get the initial path direction and the second component performs action selection to generate the final path. Zhang et al. [24] presented a deep interactive reinforcement learning method to perform the path-planning task for autonomous underwater vehicles (AUVs). The robot learns the policy from human rewards, and environmental rewards simultaneously. This twin reward policy increased the learning speed of the AUV agent.

Josef et al. [25, 26] presented a deep RL method to generate a local path in an environment with rough land. The authors used an integrated agent named Rainbow to improve the environment exploration and outperformed the deep Q-network (DQN). The researchers extended the Q-learning algorithm with an impediment area extension strategy and improved the learning speed for territory exploration. Jiang et al. [27, 28] proposed a deep reinforcement learning approach for path planning and obstacle avoidance using the strategy of experience replay and heuristic knowledge. Experience replay is performed by neural network training on experience data that is collected by the robot's movement in an unknown environment. Whereas, the heuristic function provided refined data points for the training of neural networks. The authors of [29, 30] presented a deep reinforcement learning approach using an undirected reward function for space robots. They trained an ant-like robot to wander about in an unknown environment within a defined boundary for space exploration. The authors trained agents with randomly produced route points associated with reward function using an artificial neural network (ANN). Similarly, Ruan et al. [31, 32] presented an end-to-end deep 3 Q-networks (D3QN) algorithm to perform path planning from source to destination autonomously in an unknown environment with a red, green, blue-depth (RGB-D) camera only.

Zhao et al. (2023) [33] presented a novel deep reinforcement learning (DRL) framework for dynamic robot path planning in complex environments. The proposed method integrates a multi-agent system to handle real-time obstacles and employs a DQN for efficient path optimization. The results demonstrated significant improvements in computational efficiency and path reliability compared to traditional algorithms. Kumar and Singh (2022) [34] explored the application of DRL in multi-robot systems for coordinated path planning. They introduced a policy gradient-based DRL algorithm to ensure collision avoidance and task allocation among robots. Their approach showed enhanced performance in terms of reduced path length and computation time, particularly in dense and dynamic environments. In [35], the authors address the challenge of decision-making for autonomous vehicles in the presence of obstacle occlusions, proposing the Efficient-Fully parameterized Quantile Function (E-FQF) model. Using distributional reinforcement learning, the model optimizes for worst-case scenarios, improving decision efficiency and reducing collision rates compared to conventional reinforcement learning methods. Additionally, the model shows robustness in data loss scenarios, outperforming methods with embedded long and short-term memory.

Li et al. (2024) [36] proposed a DRL-based adaptive path planning method for autonomous underwater vehicles (AUVs). By incorporating a deep deterministic policy gradient (DDPG) algorithm, the study addressed the challenges of underwater navigation, such as current dynamics and limited visibility. The experimental results indicated that the DRL approach outperformed conventional methods in terms of adaptability and robustness. Wang and Chen (2023) [37, 38] investigated the use of DRL for path planning in agricultural robots. They developed a model-free DRL approach using proximal policy optimization (PPO) to navigate robots through crop fields with minimal crop damage. Their findings highlighted the efficiency of DRL in optimizing path planning under varying environmental conditions, demonstrating potential applications in precision agriculture. Ahmed et al. (2024) [39, 40] introduced a hierarchical DRL framework for urban robot navigation. Their method leverages a combination of DQN and actor-critic algorithms to manage long-term navigation goals and short-term obstacle avoidance. The study showed that the hierarchical approach significantly improved the scalability and performance of robot path planning in urban settings.

The study [41, 42] presented a multi-constraint and multi-scale motion planning method for automated driving with the use of constrained reinforcement learning, named RLTT. The presented algorithm first generated the dynamic model of the vehicle. Later, the trajectory lane model (TLM) was formulated based on the dynamic model resulting in constrained RL actions and space states. Vehicle path space and RLTT are used to achieve macro-scale and micro-scale path planning. Lastly, the rule-based algorithm was used to generate a suitable path for autonomous vehicles. A deep deterministic policy gradient (DDPG) based path planning algorithm is presented in [43]. The presented algorithm solved the issue of slow convergence of DRL in a dynamic environment. However, navigation in a 3D environment still remains an open issue.

## Proposed approach

Reinforcement learning explicitly considers the problem of a goal-directed agent interacting with an uncertain environment. The interaction between the agent and the uncertain environment is usually formed as a Markov decision process (MDP). In general, a MDP consists of a set of states S, a set of actions A, a reward R, transition probability, and a discount factor. The proposed approach is RL based method incorporating a less complex DNN than existing approaches. The following assumptions were made for the proposed deep reinforcement learning algorithm.

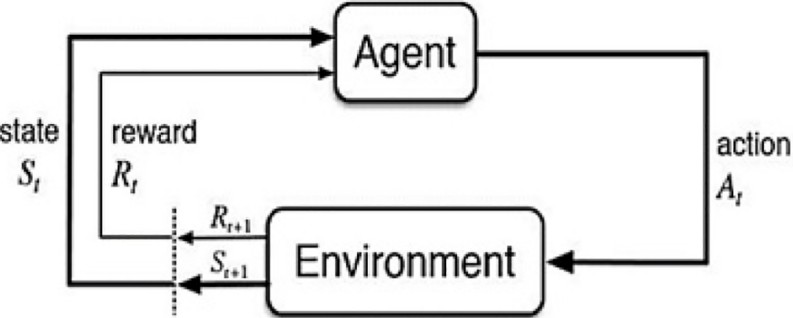

**Fig 1. The agent-environment interaction in reinforcement learning** [3].

i. The environment for the robot is represented as a 2D grid.

ii. There should be a finite number of obstacles in the environment.

iii. The environment is closed.

iv. The robot must be able to identify all the obstacles present in its vicinity.

v. The localization sensors are working fine.

The primitive concepts of the proposed approach and complete procedure are explained in subsequent sections.

## Preliminaries

**The agent.** In RL, the agent is the component that decides what action to take. The agent is free to use any observations from the environment as well as any internal rules to make the decision.

**The reward.** In RL, the agent performs some action, and after performing some action, the agent gets some reward, which is the scalar value [44]. For example, if the agent reaches the target, then the agent gets *1* value as a reward, and if the agent collides with obstacles or is not reached at the target point, then the agent gets -**1** value as a reward.

**Actions.** In RL-based algorithms, the actions can be categorized as either discrete or continuous. In discrete action, agents move up, down, left, or right, whereas, the value of action is continuous in a continuous action. Fig 1 shows the agent-environment interaction in RL. The agent acts on the environment, and the reward is awarded to the agent along with information about the next state based on the performed action.

## Q-learning

Q-learning is one of the most basic reinforcement learning algorithms. In a grid world, the simplest Q learning could solve the discrete action and state problem of finding a direction. It provides a Q table, with the rows representing discrete states and the columns representing the various acts taken at each state. The value for each grid $Q(s, a)$ denotes the future discounted reward at the state presented as s when an action is taken. If the value in the table is the ground truth, the agent should choose the best course of action based on the highest value in each column. The Q table must be initialized with an arbitrary fixed value before learning can begin. The agent then chooses an action at each time t, observes a reward presented as $r_t$, enters a new state $s_{t+1}$, and Q is modified. The algorithm is a simple value iteration update; Eq 1

represents it.

$$Q^{new}(s_t, a_t) = (1 - \alpha)Q^{odd}(s_t, a_t) + \alpha(r_t + \gamma maxQ(s_{t+1}, a)) \tag{1}$$

- $Q^{new}(s_t, a_t)$: This represents the updated Q-value for taking action $a_t$ in state $s_t$.

- $Q^{old}(s_t, a_t)$: This is the current Q-value for taking action $a_t$ in state $s_t$ before the update.

- $\alpha$: The learning rate, which determines how much new information overrides the old information. It ranges between 0 and 1. A higher $\alpha$ gives more weight to new information.

- $r_t$: The reward received after taking action $a_t$ in state $s_t$.

- $\gamma$: The discount factor, which determines the importance of future rewards. It ranges between 0 and 1. A higher $\gamma$ makes future rewards more significant.

- $maxQ(s_{t+1}, a)$: The maximum Q-value for the next state $s_{t+1}$ over all possible actions $a$. This represents the best possible future reward obtainable from the next state.

**Explanation of the equation.** The equation is a weighted average of the old Q-value and the learned value (the reward plus the discounted maximum future Q-value). Here's a step-by-step breakdown:

**i. Current Q-value (Old):**

$$(1 - \alpha)Q^{old}(s_t, a_t)$$

- This term retains a portion of the current Q-value.

- The factor $(1 - \alpha)$ reduces the influence of the old Q-value as $\alpha$ increases.

**ii. Learned Value (New Information):**

$$\alpha(r_t + \gamma \max Q(s_{t+1}, a))$$

- This term represents the new information gained from taking action $a_t$ in state $s_t$.

- The reward $r_t$ is the immediate payoff for the action.

- $\gamma maxQ(s_{t+1}, a)$ is the highest expected future reward from the next state $s_{t+1}$, discounted by $\gamma$.

**iii. Update Rule:**

- The updated Q-value $Q^{new}(s_t, a_t)$ is a combination of the old Q-value and the new information.

- The learning rate $\alpha$ controls the balance between old and new information. If $\alpha$ is close to 1, the update relies heavily on new information; if it is close to 0, the update relies more on the old Q-value.

## Environment formulation

In this study, the motion planning problem is expressed as a Markov decision process, which is represented as a combination (S, A, T, R), where S and A denote the system's state space and action space, respectively. The method takes an action a $\epsilon$ A at each time stage and transfers it from one entity to another. The possibility that the function will be in state S after taking action a is $P(s'|s, a).R: S \times A \rightarrow R$, in this method, R stands for reward function. At each time stage, the mechanism earns a real-valued reward dependent on its state $s$ and behavior $a$. The $\gamma \in$ [0, 1] is a discount factor that reflects the performance of immediate rewards over future ones.

The agent's action selection is called policy $\pi$: $S \rightarrow A$ which describes an action a $\in$ A for each state s $\in$ S. If a policy produces the highest estimated return from the initial state, it is said to be optimal. The value function v$\pi$(s) is defined as an anticipated return beginning with state s, i.e., S = s, and continuously following policy $\pi$. While the maximal viable value of v$\pi$(s) is defined as v $\times$ (s). Particularly for a state s, an action a, and a policy $\pi$, the action value of the pair (s, a) under $\pi$ is defined as q$\pi$(s, a), while the excellent action-value function is called q $\times$ (s, a). If the state transition probability P is known in a model-based environment, then dynamic programming techniques can be used to solve the MDP problem. In a model-free environment, the transition probability is unknown, which is suitable for RL.

## Action selection

The mobile robot can take four possible actions. The robot can move forward, backward, left, or right. At the same instance time $t$, the robot can take only one action, whether it moves left, right, backward, or forward.

## Reward function selection

The reward function for the mobile robot is designed as shown in Eq 2

$$R = \begin{cases} 1, & \text{Reach the target} \\ -1, & \text{Encountering obstacles} \\ 0, & \text{Otherwise} \end{cases} \quad (2)$$

If the mobile robot reaches the target, then the robot gets a positive reward; the value is one. If the mobile robot collides with obstacles, then the robot gets a negative reward; the value is -1, and the reward is zero in other circumstances.

## Step function

To combine robot motion planning with DRL, we need to transmit the problem into the framework of DRL. Fig 2 shows how the agent communicates with the environment through step( ). Therefore, we developed an agent-environment interface. To understand the DRL setting, we will start with key functions. It has step( ), reset( ), and render( ) functions. As a matter of reality, the render( ), reset( ), and step( ) decide the interactive rules that occur between the agent and the environment. Reset() and render( ) are used to reset and visualize the DRL environment. As indicated in Fig 2, through step( ), the agent interacts with the gym framework environment. After initialization, the agent chose action A to communicate with the environment.

The agent discovers whether a collision occurs on a particular move with obstacles or not, which is represented by the Boolean flag. If it is true or yes, then the environment is reset and

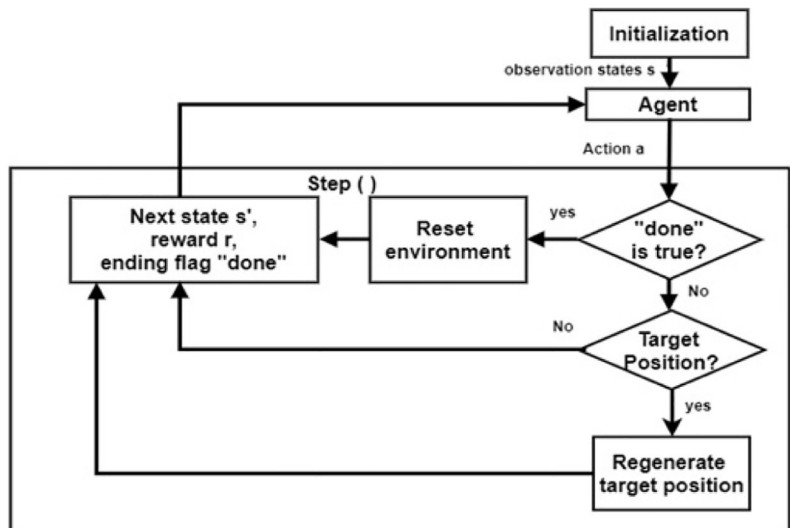

**Fig 2. Flowchart of step() function.**

returns the current reward, next state, and ending flag. If it is false, provided that the agent has moved to the target position, it regenerates another target position.

## Detailed architecture of deep reinforcement Q-learning

In the DQN framework, the neural network is employed to approximate the Q-value function $Q(s, a)$, which predicts the expected cumulative reward for taking action $a$ in state $s$. The policy is not explicitly represented by a separate neural network, as would be the case in an Actor-Critic structure. Instead, the policy is implicitly defined by the Q-value estimates, where the action selection follows an epsilon-greedy strategy: $\pi(s) = \text{argmax}_a Q(s, a)$. As shown in Fig 3, the network has two fully connected layers, with each layer containing 128 neurons. The input is 2-dimensional state information. The proposed approach transforms this information about the 2-D environment into a numerically quantifiable model to feed the neural network. This simple pre-processing step makes the overall architecture lightweight for training and testing. The input is 2-dimensional state information. The output is the action value that the agent used to move to the next state. Between the hidden layers, the ReLU activation function is used. Adam used the loss function as an optimizer function; the loss function is a mean square error.

Fig 4 shows the proposed DQN. In the proposed algorithm, we used a feed-forward neural network to predict the best Q-values. For this approach to work, the agent has to store previous experience in memory. Previously saved experience is used to train the network. Deep Q learning uses the experience-replay concept to improve performance. This concept is used to stabilize the training process.

Experience replay is nothing more than the memory that stores those experiences in the form of a tuple $(s, s', a, r)$, where $s$ is the state and $s'$ is the next state, a represents action, and $r$ is the reward. The whole process of deep Q learning is to provide the state of the environment to the network. To obtain the Q-values of all possible actions in the given state, the agent utilizes the Q-network. Based on the epsilon value and the agent's choice of random action a, the agent performs action a and observes reward r and the next state $s'$. This information is stored in the experience replay memory $(s, s', a, r)$. Then a sample of random batches from experience

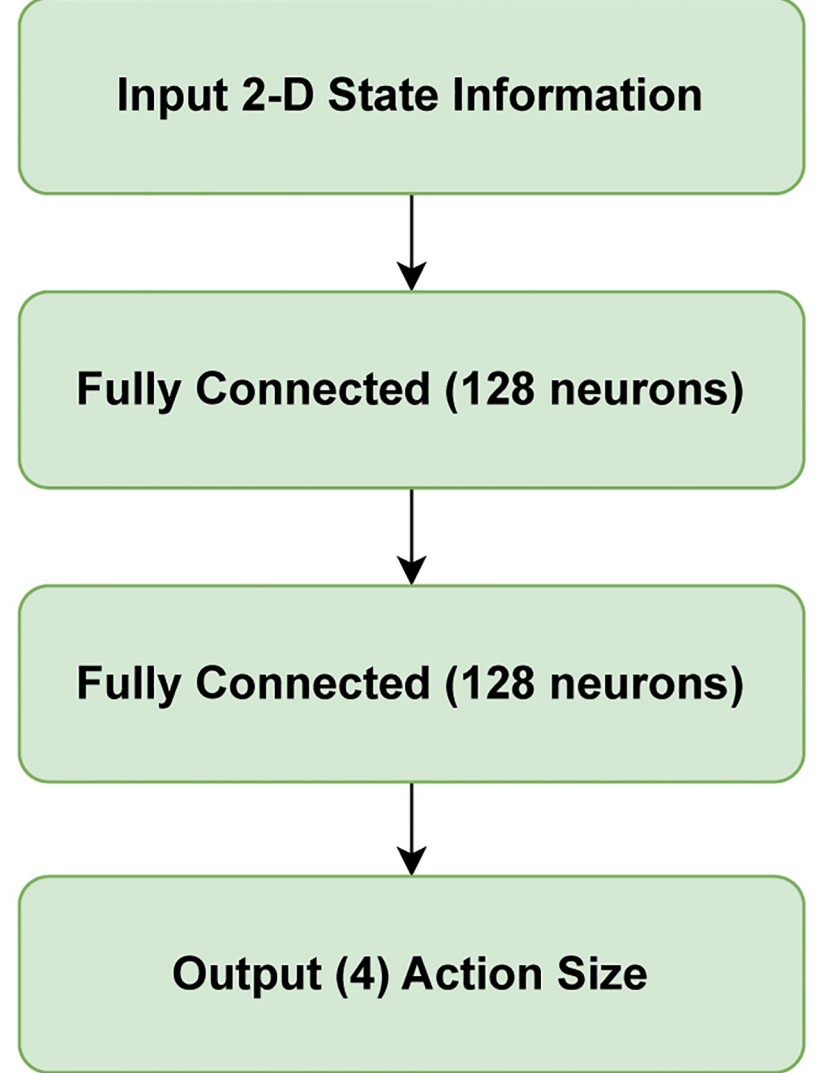

**Fig 3. Network architecture.**

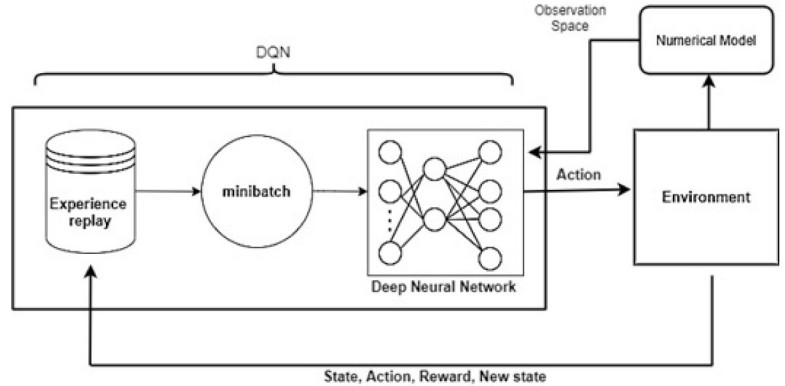

**Fig 4. The main components and data flow of the proposed algorithm.**

**Table 1. Summary of neural network architecture.**

| Layer(Type) | Output Shape | No. of Parameters |
|---|---|---|
| flatten (Flatten) | (None, 121) | 0 |
| dense (Dense) | (None, 128) | 15616 |
| dense_1 (Dense) | (None, 128) | 16512 |
| dense_2 (Dense) | (None, 4) | 516 |

replay memory was used to perform training on the Q-Network. Table 1 shows the summary of the neural network. In the neural network, the total trainable parameters are 32,644.

The pseudo-code for the proposed algorithm with experience replay is presented in algorithm 1.

**Algorithm 1** Novel Deep Reinforcement Learning Based Collision Avoidance Approach for Path Planning of Robots in Unknown Environment.

```
 1: Input: Environment map coordinates of Source and Goal positions
 2: Initialize: Experience replay memory D to capacity N
 3: Initialize: Action-value function Q with random weights θ
 4: Initialize: Target action-value function Q̂ with weights θ⁻ = θ
 5: Set: Exploration rate ϵ ∈ [0, 1], decay rate ϵ_decay, minimum ϵ_min
 6: Set: Discount factor γ ∈ [0, 1], learning rate α
 7: for episode = 1 to M do
 8:   Initialize environment and receive initial state s₁
 9:   for t = 1 to T do
10:     With probability ϵ select a random action aₜ
11:     Otherwise select aₜ = argmaxₐQ(sₜ, a;θ)
12:     Execute action aₜ and observe reward rₜ and next state sₜ₊₁
13:     Store transition (sₜ, aₜ, rₜ, sₜ₊₁) in D
14:     Sample random mini-batch of transitions (sⱼ, aⱼ, rⱼ, sⱼ₊₁) from D
15:     Compute:
```

$$Y_j = \begin{cases} r_j & \text{if terminal state} \\ r_j + \gamma \max_{a'} \hat{Q}(s_{j+1}, a'; \theta^-) & \text{otherwise} \end{cases}$$

```
16:     Perform gradient descent step on (yⱼ - Q(sⱼ, aⱼ;θ))² with
        respect to the network parameters θ
17:     Every C steps, update Q̂ = Q (i.e., θ⁻ = θ)
18:     Set sₜ = sₜ₊₁
19:     Decrease ϵ by ϵ_decay, until ϵ_min is reached
20:     if reached goal state or maximum steps then
21:       End episode
22:     end if
23:   end for
24: end for
```

## Results and discussion

### Experimental setup

The algorithms are implemented using Python 3 and tested on a PC with an Intel i3–400M @ 2.40 GHz CPU and 4 GB internal RAM. The operating system is 64-bit Windows 10. Two different environment map case studies are adopted from benchmark data sets [6, 45]. A software simulator is built for the experimentation of the mobile robot agent working in the

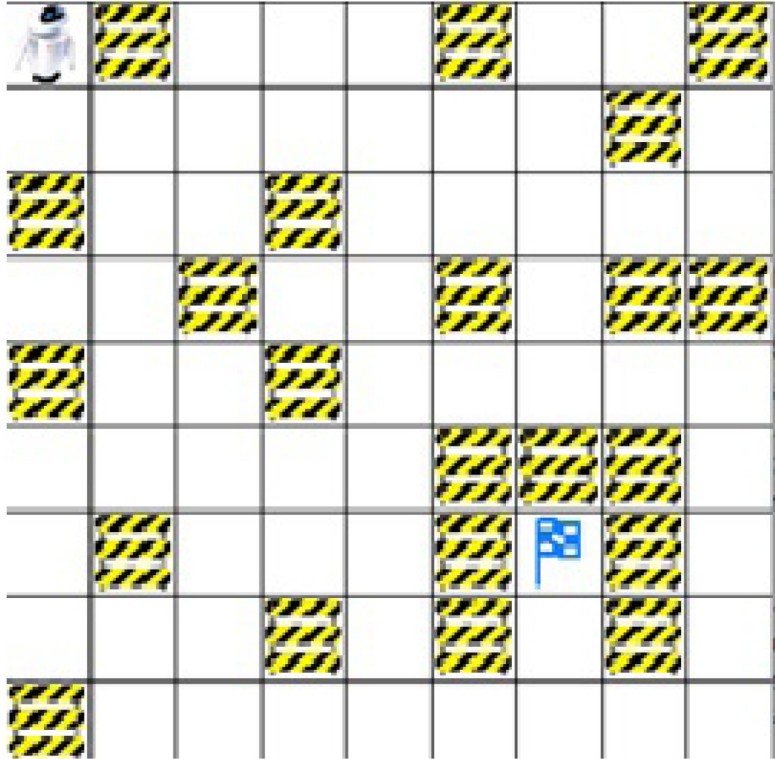

**Fig 5. Map M1: Environment cluttered with structured obstacles.**

2-dimensional environment to ensure the efficient performance of the proposed approach. The 2-D environment is built in the openAI gym framework [46]. It is an open-source framework used for designing the custom environment and comparing the RL algorithm.

## Case studies environment

Experiments in simulation show an indoor environment. The environment is a grid world with a size of 9 × 9. M1 is the scenario of an environment cluttered with obstacles. In M1, the agent's initial position is in the top left corner, and its target position is flagged as shown in Fig 5.

Another environment M2 is a complex, narrow passage with an occupancy grid size of 11 × 11. In the M2 environment agent, the initial position is the top upper left corner, and it has to find a target, which is the bottom right corner, as shown in Fig 6.

As shown in the environment maps, the initial position of a mobile robot is in the upper left corner. The environment is grid-based, and obstacles are shown by occupied grid cells in the occupancy grid environment map. In the environment, the task of the mobile agent is to find the path from the initial position to a target position shown in the environment by the flag. The mobile agent can move in free spaces, as shown by white grid cells. The environment is unknown to the agent.

## Evaluation measurement

The following evaluation metrics are used to assess the proposed approach's outcome and results.

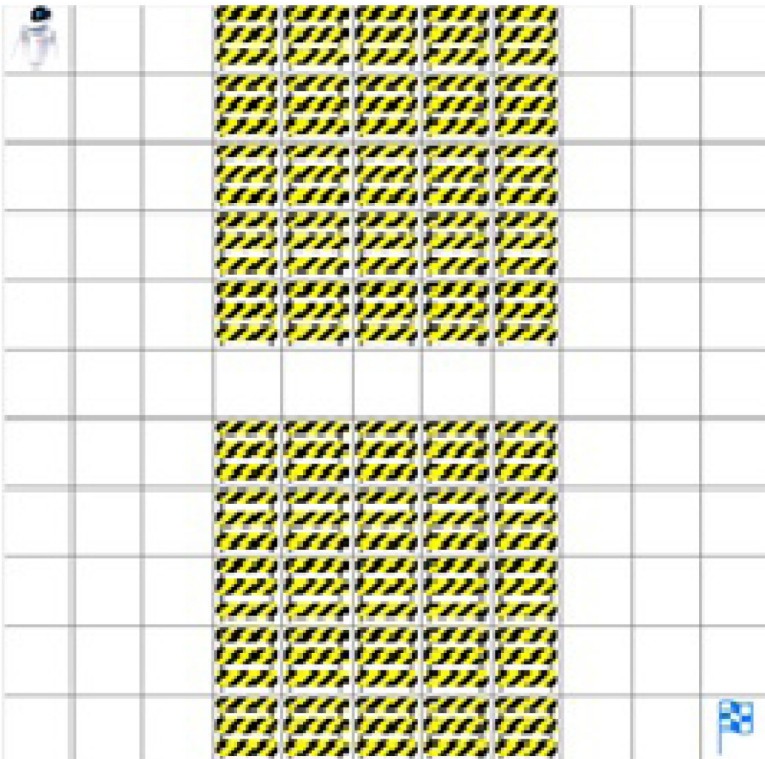

**Fig 6. Map M2: Environment showing obstacles in formation of a narrows passage.**

i. **Number of convergence episodes** means how many episodes the agent found the path to the target position. Convergence with a smaller number of episodes is desirable. Moreover, an episode could have a greater number of steps but may be better than another episode due to the success rate of collision avoidance and the identified position score. Hence, a good episode is not dependent on a number of steps, but it is useful based on the success rate.

ii. **Number of turns** means how many turns the robot takes to reach a target position. It is an important metric for motion planning. If a path has a greater number of turns, it will require the robot to first decelerate at turns, reorient its angle and direction, and then accelerate again. Consequently, this whole process will not only consume more path following time during application but will also consume more energy for the robot and even cause exertion for the robot's controller. Moreover, a path offering unnecessary turns also causes the robot's premature aging.

## Results

This section provides a discussion of the experiment results for the proposed approach in two different environments. Each environment has different obstacles. The mobile robot moves in the environment to do motion planning from its initial position to the target point. A number of experiments were conducted to find out the best values for the hyperparameters of the algorithm. Simulations are performed to compare the effects of different learning rates, exploration

**Table 2. Final parameter values for the proposed approach after trend analysis.**

| Exploration factor | Learning rate | Discount Factor | Activation Function | Optimizer | Memory | Batch Size |
|---|---|---|---|---|---|---|
| 0.9 | 0.01 | 0.95 | ReLU | ADAM | 20000 | 32 |

**Table 3. Trend analysis parameter's impact for M1.**

| Discount factor | Exploration factor | Learning rate | No. of convergence episodes | Maximum steps | Minimum steps |
|---|---|---|---|---|---|
| 0.4 | 0.5 | 0.4 | 650 | 110 | 17 |
| 0.5 | 0.6 | 0.3 | 600 | 140 | 16 |
| 0.6 | 0.7 | 0.2 | No convergence | 320 | 15 |
| 0.7 | 0.8 | 0.1 | No convergence | 300 | 17 |
| 0.8 | 0.85 | 0.01 | No convergence | 500 | 16 |

factors, and discount factors on the performance of an algorithm. We tested the proposed approach in three aspects: the number of convergence episodes, the number of turns, and the path length. Table 2 demonstrates the finalized success parameters of the proposed approach.

Tables 3 and 4 show the results from different parameters for M1 and M2. It is obvious from the given results that the learning rate and exploration factor have a greater impact on the performance of the proposed approach. The impact parameters used in Tables 3 and 4 are described as follows:

i. **Discount rate** tells how important future rewards are to the current state. The discount factor has a value between 0 and 1. As shown in Tables 3 and 4, as we increase the value of the discount rate, the algorithm converges.

ii. **Learning rate** is a configurable hyperparameter used in the training of neural networks that has a small positive value, often in the range between 0.0 and 1.0.

iii. **Exploration factor** is a configurable parameter. The agent used the exploration factor to explore the environment. Its value is between 0 and 1. If the value is near 0, the agent does not explore the environment, and if the value is near 1, the agent is exploring the environment. As shown in Tables 3 and 4, as the exploration factor increases, the agent finds the path to reach the target point.

## Comparison with existing approach

Map M1 is the scenario of the structured cluttered environment with obstacles shown in Fig 7a. The proposed deep Q-learning algorithm showed better results than the Q-learning

**Table 4. Trend analysis parameter's impact for M2.**

| Discount factor | Exploration factor | Learning rate | No. of convergence episodes | Maximum steps | Minimum steps |
|---|---|---|---|---|---|
| 0.6 | 0.6 | 0.01 | No convergence | 650 | 20 |
| 0.7 | 0.7 | 0.1 | No convergence | 700 | 19 |
| 0.8 | 0.8 | 0.2 | 700 | 800 | 19 |
| 0.85 | 0.88 | 0.3 | 680 | 600 | 20 |
| 0.9 | 0.9 | 0.4 | 650 | 300 | 21 |

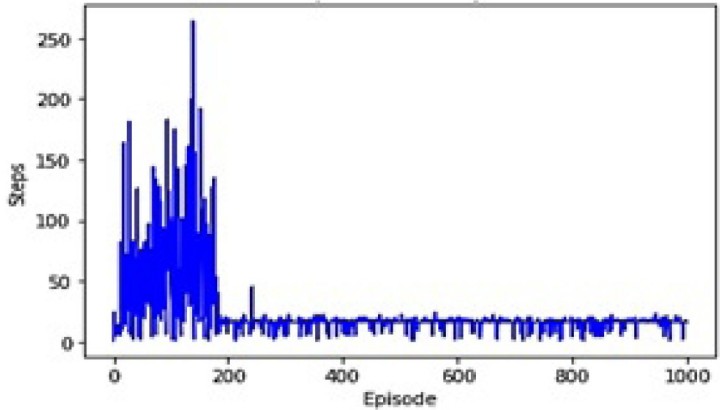

**(a)**

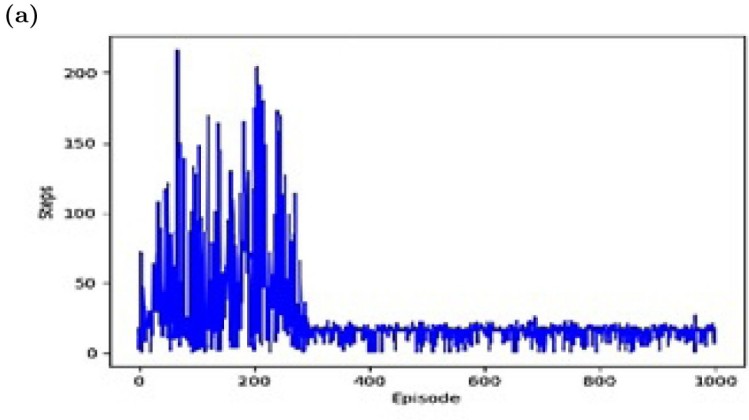

**(b)**

**Fig 7. Episodes via steps for M1.** (a) Proposed approach. (b) DQN approach [45].

algorithm. It is obvious by the plots shown in Fig 7b that using the previous Q-learning agent, the path to the target was found in 300th episode and it took quite more episodes to stabilize. However, the path to the target was found in 210th episodes using the proposed approach agent, as shown in Fig 7.

Fig 8 shows the path planning of the proposed approach and existing Q-learning approach for environment map case M1. It is seen from simulation results that our approach for motion planning for the robot is better than the existing approach in M1 because agents learn better actions to achieve the target.

The proposed deep Q-learning algorithm showed better results than the Q-learning algorithm. It is obvious by the plots shown in Fig 9 that using the previous Q-learning agent, the path to the target was found in 650th episode and it took quite more episodes to stabilize. However, the path to the target was found in 400th episodes using the proposed approach agent, as shown in Fig 9.

Fig 10 shows the path planning of the proposed approach and the path planning of the existing approach for a robot in M2. It is observed in simulation results that the proposed approach for motion planning for the mobile robot outperformed the existing Q-learning approach in M2 as well. There are also other deep reinforcement learning approaches as well however, they are resource and computing-intensive and use complex deep neural networks

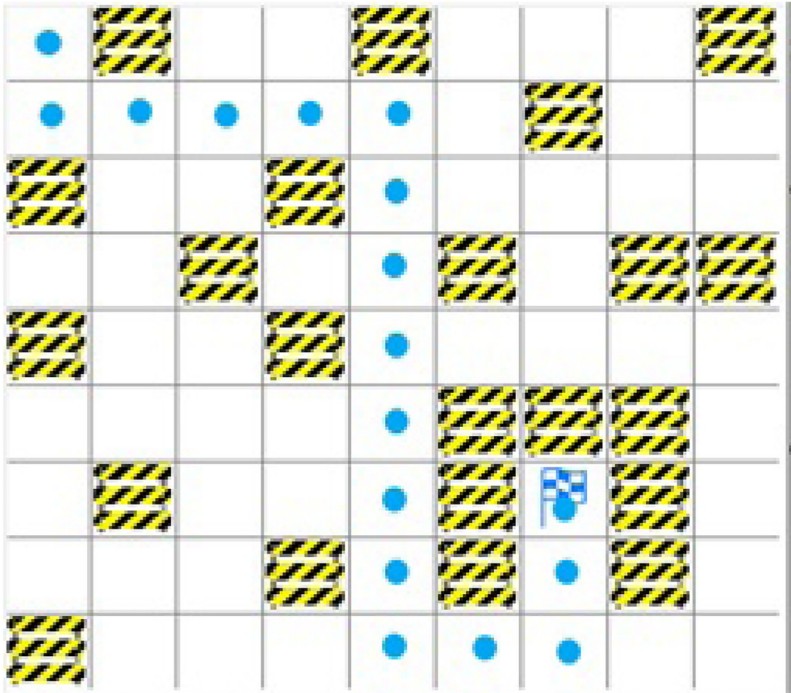

(a)

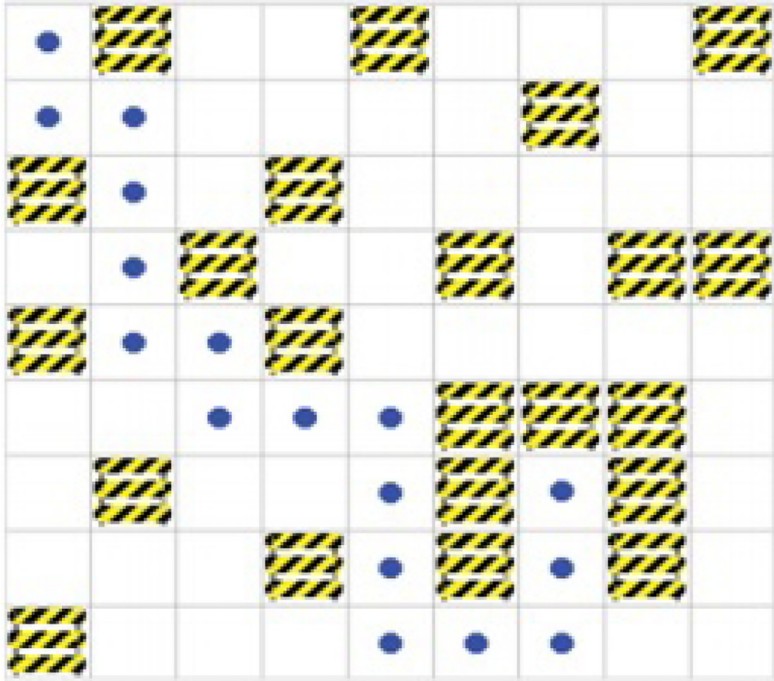

(b)

**Fig 8. Path planned for environment Map M1.** (a) Proposed approach. (b) DQN approach [45].

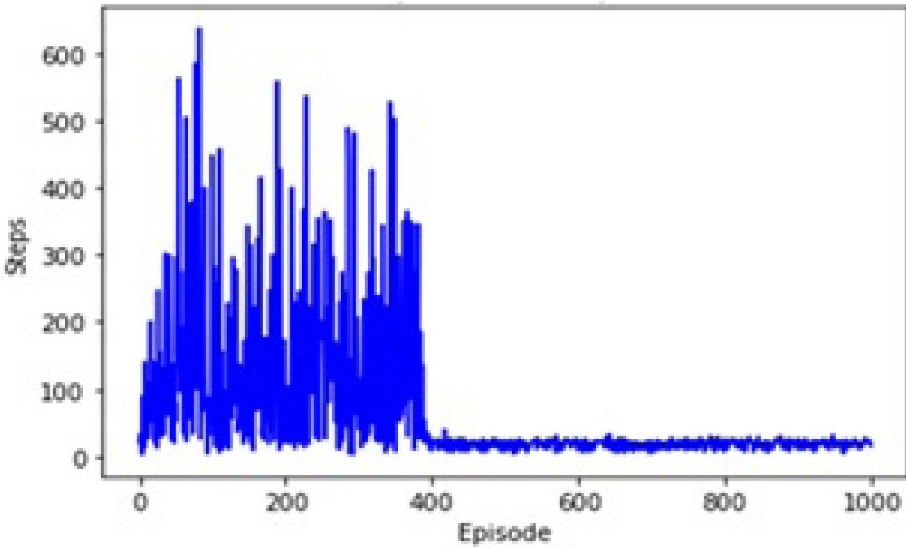

(a)

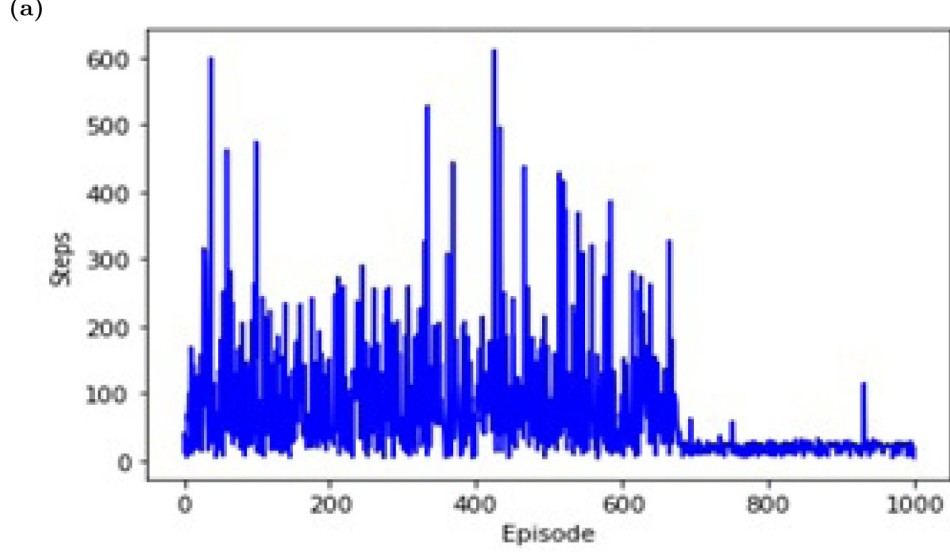

(b)

**Fig 9. Episodes via steps for M2.** (a) Proposed approach. (b) DQN approach [45].

while working in 3-D mode. Whereas, the proposed approach offers a relatively lightweight deep net operating in a 2-D environment and has improved convergence significantly.

Table 5 shows the comparative analysis of the proposed approach with the existing approach according to evaluation matrices for M1. Results show the comparative analysis for M1 and M2 of our proposed approach with the existing approach according to evaluation matrices. In the previous Q-learning approach, the agent maintains a table with the states as rows and actions as columns. The agent chooses an action from the table to go next state and a reward is given to the agent for each action it chooses. Using the table the agent interacts with the environment and measures the reward for each action.

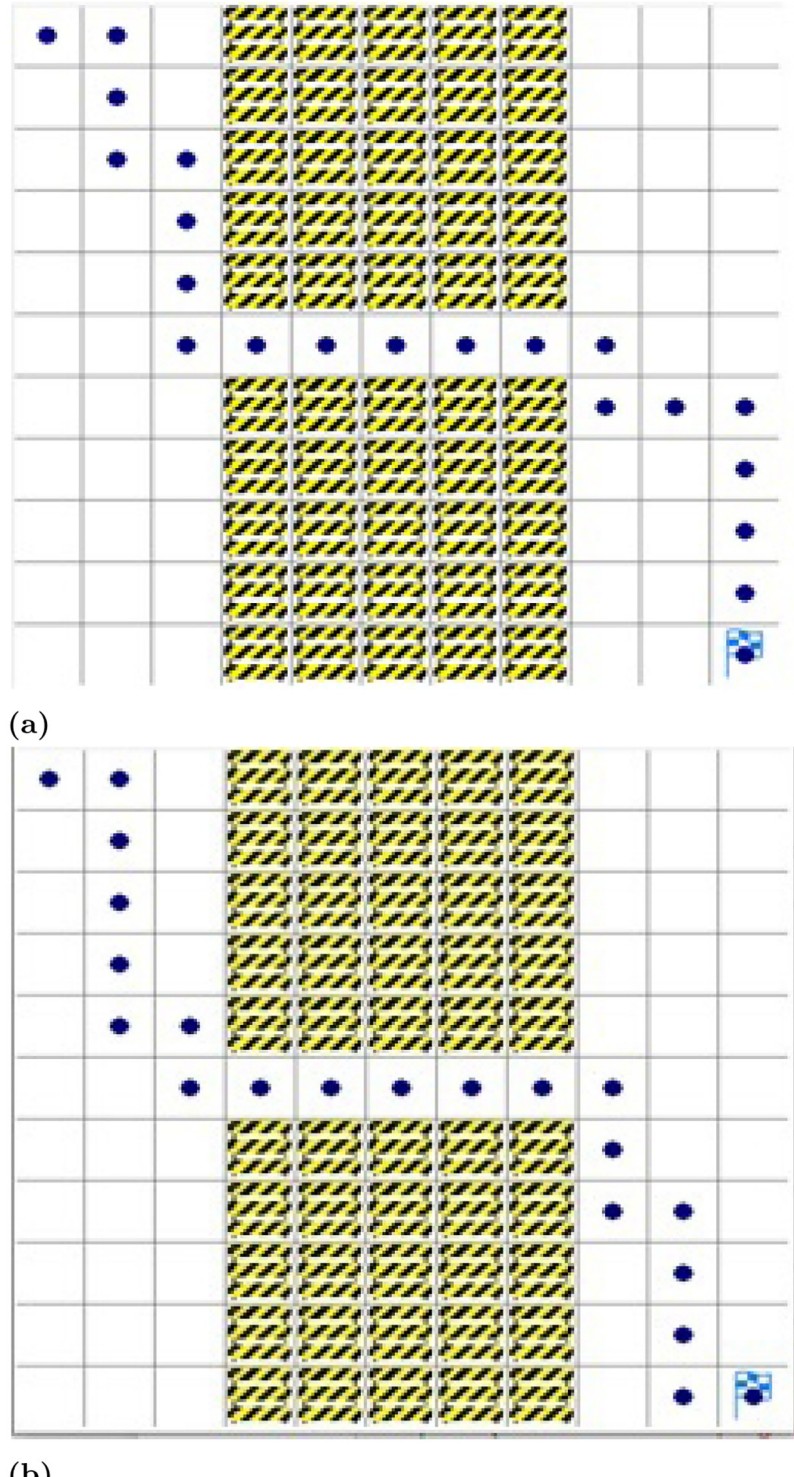

**Fig 10. Path planned for environment map M2.** (a) Proposed approach. (b) DQN approach [45].

**Table 5. Comparison of a proposed and existing approach [45].**

| Environment Map | Evaluation Matrices | Proposed Approach | Traditional DQN [45] |
|---|---|---|---|
| M1 | Convergence episodes | 210 | 300 |
| | No. of turns | 3 | 7 |
| M2 | Convergence episodes | 400 | 650 |
| | No. of turns | 6 | 7 |

The proposed approach has used a deep neural network for going to the next state instead of the table. The input of the neural network is observation space which is a 2-D array and the output of the neural network is action space which is action size, based on the epsilon value the agent chooses a random action and after the agent performs the action it ob-serves reward and next state. In our proposed approach we utilize the concept of experience replay memory and mini-batch. Experience replay is nothing more than a memory that stores the experience of the agent. We choose random samples from experience replay memory to perform the training of neural network which is the concept of mini-batch.

## Significance of the proposed algorithm compared to traditional DQN

To effectively address novality of the proposed algorithm compared to the traditional Deep Q-Network (DQN) algorithm, we highlight the following points:

i. **Simplified Architecture for Motion Planning:**

- The proposed approach uses a less complex deep reinforcement learning model specifically tailored for motion planning in environments cluttered with obstacles. This reduces computational complexity and improves efficiency in training and execution compared to traditional DQN, which may not be optimized for such specific tasks.

ii. **Enhanced Environment Handling:**

- The algorithm is tested in two types of challenging environments: narrow, cluttered pas-sages and environments with structural obstacles. The traditional DQN algorithm is gener-ally tested in simpler, grid-based environments. This testing in more complex scenarios demonstrates the robustness and adaptability of the proposed method.

iii. **Improved Convergence and Path Efficiency:**

- The proposed approach achieves faster convergence to an optimal policy by focusing on creating paths with fewer turns. This not only reduces the time taken to find the path but also minimizes energy consumption, leading to more efficient and practical solutions for real-world robotic applications.

iv. **Experience Replay Optimization:**

- The experience replay mechanism is fine-tuned to better handle the diverse and dynamic nature of the environment, ensuring that the learning process is more stable and effective. This optimization leads to better generalization and performance of the algorithm com-pared to the standard experience replay used in traditional DQNs.

v. **Energy-Efficient Path Planning:**

- The focus on generating an energy-efficient path with fewer turns distinguishes the proposed approach from traditional DQN algorithms. This innovation is particularly significant in robotics, where energy efficiency directly impacts the operational lifespan and practicality of autonomous systems.

## Conclusions

This study investigates deep reinforcement learning algorithms for the task of motion planning in a given environment. A reinforcement learning approach is proposed with a relatively less complex deep neural network. It takes input in a 2D array format and operates on the principle of feed-forward. Experiments are carried out in both cluttered and narrow passage 2D environments with static obstacles. The performance of the proposed approach is evaluated based on exploration factor 0.9, discount factor 0.95, and learning rate 0.01 parameters. The experimental results demonstrate that the proposed approach is stable and converges faster than the existing approaches. Results show that using the proposed approach, the agent finds a path in 210 episodes of training in a cluttered environment, and the agent finds a path in 400 episodes of training in a narrow passage environment. Moreover, the planned path contains less number of possible turns which enhances its energy efficiency as compared to the previous approach. Future work for this study is to explore motion-planning tasks for robots in a 3D environment with dynamic obstacles. Moreover, working with continuous action is also a future direction for this study. Integration of sampling-based methods such as RRT*-AB with the proposed reinforcement approach and its extension for the area coverage problem is another challenging future work dimension.

## Supporting information

**S1 File.**
(PDF)

## Author Contributions

**Conceptualization:** Raed Alharthi, Turki Aljrees, Nisreen Innab.

**Data curation:** Raed Alharthi, Iram Noreen, Amna Khan.

**Formal analysis:** Zoraiz Riaz.

**Funding acquisition:** Raed Alharthi, Turki Aljrees, Nisreen Innab.

**Investigation:** Amna Khan, Turki Aljrees, Zoraiz Riaz, Nisreen Innab.

**Methodology:** Iram Noreen, Amna Khan, Zoraiz Riaz.

**Project administration:** Raed Alharthi, Turki Aljrees.

**Resources:** Iram Noreen, Amna Khan, Nisreen Innab.

**Software:** Iram Noreen, Zoraiz Riaz.

**Supervision:** Turki Aljrees.

**Validation:** Iram Noreen, Amna Khan, Turki Aljrees, Zoraiz Riaz.

**Visualization:** Raed Alharthi, Amna Khan, Zoraiz Riaz.

**Writing – original draft:** Iram Noreen, Amna Khan, Zoraiz Riaz.

**Writing – review & editing:** Raed Alharthi, Turki Aljrees, Nisreen Innab.

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
