## [Decision Letter · Decision Letter 0]

4 Jun 2024

PONE-D-24-20036Novel Deep Reinforcement Learning Based Collision Avoidance Approach for Path Planning of Robots in Unknown EnvironmentPLOS ONE

Dear Dr. Aljrees,

Thank you for submitting your manuscript to PLOS ONE. After careful consideration, we feel that it has merit but does not fully meet PLOS ONE’s publication criteria as it currently stands. Therefore, we invite you to submit a revised version of the manuscript that addresses the points raised during the review process.

Please submit your revised manuscript by Jul 19 2024 11:59PM. If you will need more time than this to complete your revisions, please reply to this message or contact the journal office at plosone@plos.org. Please include the following items when submitting your revised manuscript:A rebuttal letter that responds to each point raised by the academic editor and reviewer(s). You should upload this letter as a separate file labeled 'Response to Reviewers'.A marked-up copy of your manuscript that highlights changes made to the original version. You should upload this as a separate file labeled 'Revised Manuscript with Track Changes'.An unmarked version of your revised paper without tracked changes. You should upload this as a separate file labeled 'Manuscript'.

We look forward to receiving your revised manuscript.

Kind regards,

Lei Zhang, PhD

Academic Editor

PLOS ONE

Journal Requirements:

**Additional Editor Comments:**

However, there are many technological issues that need to be addressed or clarified before qualifying for this journal publication.

Reviewers' comments:

Reviewer's Responses to Questions

**Comments to the Author**

1. Is the manuscript technically sound, and do the data support the conclusions?

Reviewer #1: Partly

Reviewer #2: Partly

2. Has the statistical analysis been performed appropriately and rigorously? 

Reviewer #1: N/A

Reviewer #2: No

3. Have the authors made all data underlying the findings in their manuscript fully available?

Reviewer #1: Yes

Reviewer #2: No

4. Is the manuscript presented in an intelligible fashion and written in standard English?

Reviewer #1: No

Reviewer #2: No

5. Review Comments to the Author

Reviewer #1: This research presents a novel Q-learning-based reinforcement algorithm with deep learning integration. Generally, the manuscript is well-organized and the experiments results can well support the proposed method. I still have some suggestions and questions:

1. In the related work section, emphasis should be placed on the literature review of deep reinforcement learning research in robot path planning, and please simplify some of the less relevant parts

2. Please correct words and grammatical errors in the article, e.g. "robot?s" in line 99, and check the whole article.

3. For the first time in the text, please use the full name, such as DQN

4. In line 248, the pseudo-code for the proposed algorithm with experience replay is not represented in the paper.

5. Please explain in detail what are the innovations of the algorithm proposed in this paper compared to the traditional DQN algorithm?

6. There is a problem with the layout of Table 2, Table 3 and Table 4. There are many format problems in this article. Please check carefully.

7. The environment is a grid world with a size of 9*9, what is the basis? If applied to practical engineering, how to design the grid size occupied by obstacles?

8. the method of comparison in this paper is the Q-learning method in 2016, which is relatively old. Please highlight your own advantages for the relatively new method, which is more convincing.

9. In conclusions, you express the planned path contains less number of possible turns which enhances its energy efficiency as compared to the previous approach, but the algorithm proposed in this paper is not an optimal path in M2, how to consider this problem?

Reviewer #2: The paper claims to propose a Novel Deep Reinforcement Learning method, but there are serious problems and errors in the title, theme, methodology, and three contributions of the paper. In addition, the article also has a lot of problems in expression, image, and formula. The experimental scenario and baseline of this paper are also unreasonable. The authenticity and effectiveness of this method are also questionable.

1. In introduction, it is said that “However, current motion planning approaches lack support for timely automated responses to the environment in real-time navigation scenarios”, but can't D * be used to handle unknown and dynamic environments?

2. Ref [13] mainly describes the development process of autonomous vehicles, with the aim of learning complex driving behaviors and expert driving experience. I couldn't find any description of "grid based, sampling-based, or heuristic-evolutionary planners are slow, computationally expensive, and less responsive to the environment"

3. In introduction, it is said that “The mobile robot must follow a simple trajectory consisting of straight lines and circles to ensure simplicity”. This statement is incorrect. Simplifying the robot trajectory into straight lines and circles is a hypothesis to verify the algorithm's online and offline exploration performance.

4. In the first contribution, what is “less complex DRL”? Simple neural networks are difficult to handle complex problems and achieve high adaptability. Why use "less complex DRL" here?

5. In the first contribution, almost no difference can be found between proposed method and DQN.

6. In the second contribution, each environment only contains one obstacle map, and the map is static. It is unreasonable. The test results may also be caused by accident, which cannot prove the progressiveness of the algorithm.

7. In the third contribution, if you want to reduce energy consumption, you should start from the motor power map, and achieve energy-saving by controlling the motor torque. The description of this contribution is completely wrong.

8. In the last part of related work, it is said that “However, navigation in a 3D environment still remains an open issue”. But in Proposed Approach, it is said that “The environment for the robot is represented as a 2D grid”. The method proposed in the paper still does not solve the problem raised at the end of related work.

9. Missing explanation for variables in formula (1).

10. In the Proposed Approach, it is said that “The value function vπ(s) is … which approximates “how sweet” in that state”. The value function has a more rigorous definition and explanation, please do not use such vague and imprecise statements.

11. Please use superscripts, subscripts, and symbols correctly. Modify the variables likes “v×(s)”.

12. The reward function is too simple and its effectiveness is questionable.

13. In Detailed Architecture, it is said that “A neural network is used to describe the policy and value functions”. Is the policy using a neural network too? If the policy also uses a network, it should belong to the Actor critic structure, not DQN.

14. The specific expression for the input state is still not clear. If the input state includes obstacle information? If it does not contain any obstacle information, the network has no environmental adaptability.

15. What is "fully connected dense layers"? This seems to be a non-standard expression. Are you trying to express "fully connected layers"?

16. In fig. 2, part of the text in the image is too small and some of the text is blurry. It is recommended to redraw the image.

17. In fig. 3, the text in the image is deformed and blurred. If "Dense" refers to a fully connected layer, it is recommended to change the "Dense" in the image to "FC", which is a more general expression.

18. In Fig. 4, part of the text in the image is too small and some of the text is blurry. It is recommended to redraw the image.

19. In table 1, What does "None" mean?

20. “The pseudo-code for the proposed algorithm with experience replay is presented in algorithm ??.” We can not find the pseudocode, and there is also a formatting error in this sentence.

21. In Case studies environment, the scene settings are too simple and can be solved using methods such as dynamic programming and A *. The experimental scenario set cannot reflect the advantages of dealing with scenarios in real time mentioned in the introduction.

22. In the Evaluation measurement, how do the method achieve an improvement in turning number? In the reward function, there is even no any items related to the turning number. The authenticity and effectiveness of this method are questionable.

23. It is recommended to compare with algorithms such as DQN, TD3, SAC, etc. Comparing with Q-learning is meaningless.

6. PLOS authors have the option to publish the peer review history of their article (what does this mean?). If published, this will include your full peer review and any attached files.

Reviewer #1: No

Reviewer #2: No

---

## [Decision Letter · Decision Letter 1]

30 Aug 2024

PONE-D-24-20036R1Novel Deep Reinforcement Learning Based Collision Avoidance Approach for Path Planning of Robots in Unknown EnvironmentPLOS ONE

Dear Dr. Aljrees,

Thank you for submitting your manuscript to PLOS ONE. After careful consideration, we feel that it has merit but does not fully meet PLOS ONE’s publication criteria as it currently stands. Therefore, we invite you to submit a revised version of the manuscript that addresses the points raised during the review process.

We look forward to receiving your revised manuscript.

Kind regards,

Lei Zhang, PhD

Academic Editor

PLOS ONE

Additional Editor Comments:

The manuscript needs further revisions before qualifying for this journal publication.

Reviewers' comments:

Reviewer's Responses to Questions

**Comments to the Author**

1. If the authors have adequately addressed your comments raised in a previous round of review and you feel that this manuscript is now acceptable for publication, you may indicate that here to bypass the “Comments to the Author” section, enter your conflict of interest statement in the “Confidential to Editor” section, and submit your "Accept" recommendation.

Reviewer #1: (No Response)

Reviewer #2: (No Response)

2. Is the manuscript technically sound, and do the data support the conclusions?

Reviewer #1: Partly

Reviewer #2: Partly

3. Has the statistical analysis been performed appropriately and rigorously? 

Reviewer #1: Yes

Reviewer #2: No

4. Have the authors made all data underlying the findings in their manuscript fully available?

Reviewer #1: Yes

Reviewer #2: Yes

5. Is the manuscript presented in an intelligible fashion and written in standard English?

Reviewer #1: Yes

Reviewer #2: Yes

6. Review Comments to the Author

Reviewer #1: 1.The cited references do not summarize their problems and cannot highlight the necessity and urgency of your research.

2.What is the innovation of algorithm 2?

3. How to prove the improvement of training efficiency compared with the traditional DQN algorithm? Please add a group of comparisons. Other advantages are similar. Please explain with the comparison results, otherwise it is not convincing.

4. It is recommended not to compare Q-learning, which is not persuasive.

Reviewer #2: Some of the issues in this article have been modified, but there are still some key issues that need further modification.

1. It is suggested to cite the paper “Decision making Models on Perceived Uncertainty with Distributed Reinforcement Learning” .

2. As for the comment about “the algorithm proposed in this paper is not an optimal path in M2”, this problem must be solved, in Fig. 10, the result is clearly not the optimal solution. You should at least obtain the optimal solution in the given environment, otherwise it cannot prove the correctness and effectiveness of the algorithm proposed in this paper.

3. As for the comment “each environment only contains one obstacle map, and the map is static. It is unreasonable”, designing a dynamic or stochastic environment may be too difficult. But in addition to “the steps we plan to take to address it”, you should at least conduct experiments in some slightly more complex environments in this paper. You can refer to “An Online POMDP Solver for Uncertainty Planning in Dynamic Environment”.

4. As for the comment “The specific expression for the input state is still not clear”, this article has not been modified in any way. This is necessary, please explain this in the paper.

5. As for the comment about “fully connected dense layers”, please also modify the “dense” in Fig. 3 to “Fully Connected”.

6. As for the comment about “In the reward function, there is even no any items related to the turning number”, the explanation in the reply letter is insufficient. It is said that “the overall optimization process of the deep reinforcement learning algorithm tends to favor more direct and efficient paths, which naturally results in fewer turns”, the deep reinforcement learning algorithm tends to favor the path which gets more reward in the reward function. This reward function is slightly useful for finding direct and efficient paths because of the discount factor. However, the reward function may not have resulted in the optimal path in Fig. 10, due to its simplicity. It is suggested to changing the reward function and do experiment again.

7. As for the comment about “It is recommended to compare with algorithms such as DQN, TD3, SAC, etc”, in addition to “the steps we plan to take to address it”, you should at least use one method as baseline method in this paper.

7. PLOS authors have the option to publish the peer review history of their article (what does this mean?). If published, this will include your full peer review and any attached files.

Reviewer #1: No

Reviewer #2: No

---

## [Author Response · Author response to Decision Letter 1]

3 Oct 2024

We have provided a separate PDF to address reviewers concern.

---

## [Decision Letter · Decision Letter 2]

8 Oct 2024

Novel Deep Reinforcement Learning Based Collision Avoidance Approach for Path Planning of Robots in Unknown Environment

PONE-D-24-20036R2

Dear Dr. Aljrees,

We’re pleased to inform you that your manuscript has been judged scientifically suitable for publication and will be formally accepted for publication once it meets all outstanding technical requirements.

Kind regards,

Lei Zhang, PhD

Academic Editor

PLOS ONE

Additional Editor Comments (optional):

The revised manuscript can be accepted for publication.

Reviewers' comments:

Reviewer's Responses to Questions

**Comments to the Author**

1. If the authors have adequately addressed your comments raised in a previous round of review and you feel that this manuscript is now acceptable for publication, you may indicate that here to bypass the “Comments to the Author” section, enter your conflict of interest statement in the “Confidential to Editor” section, and submit your "Accept" recommendation.

Reviewer #1: All comments have been addressed

Reviewer #2: (No Response)

2. Is the manuscript technically sound, and do the data support the conclusions?

Reviewer #1: Yes

Reviewer #2: Yes

3. Has the statistical analysis been performed appropriately and rigorously? 

Reviewer #1: Yes

Reviewer #2: Yes

4. Have the authors made all data underlying the findings in their manuscript fully available?

Reviewer #1: No

Reviewer #2: (No Response)

5. Is the manuscript presented in an intelligible fashion and written in standard English?

Reviewer #1: (No Response)

Reviewer #2: Yes

6. Review Comments to the Author

Reviewer #1: This research presents a novel Q-learning-based reinforcement algorithm with deep learning integration. The

proposed approach is evaluated in a narrow and cluttered passage environment. The results of the experiment are reasonable, and the suggestions were responded to and addressed, so the paper is recommended for publication.

Reviewer #2: (No Response)

7. PLOS authors have the option to publish the peer review history of their article (what does this mean?). If published, this will include your full peer review and any attached files.

Reviewer #1: No

Reviewer #2: No

---

## [Editor Report · Acceptance letter]

15 Oct 2024

PONE-D-24-20036R2 

PLOS ONE

Dear Dr. Aljrees, 

I'm pleased to inform you that your manuscript has been deemed suitable for publication in PLOS ONE. Congratulations! Your manuscript is now being handed over to our production team.

Kind regards, 

on behalf of

Dr. Lei Zhang 

Academic Editor

PLOS ONE